# MODEL-BASED IMITATION LEARNING FROM STATE TRAJECTORIES

## ABSTRACT

Imitation learning from demonstrations usually relies on learning a policy from trajectories of optimal states and actions. However, in real life expert demonstrations, often the action information is missing and only state trajectories are available. We present a model-based imitation learning method that can learn environment-specific optimal actions only from expert state trajectories. Our proposed method starts with a model-free reinforcement learning algorithm with a heuristic reward signal to sample environment dynamics, which is then used to train the state-transition probability. Subsequently, we learn the optimal actions from expert state trajectories by supervised learning, while back-propagating the error gradients through the modeled environment dynamics. Experimental evaluations show that our proposed method successfully achieves performance similar to (state, action) trajectory-based traditional imitation learning methods even in the absence of action information, with much fewer iterations compared to conventional model-free reinforcement learning methods. We also demonstrate that our method can learn to act from only video demonstrations of expert agent for simple games and can learn to achieve desired performance in less number of iterations.

## 1 INTRODUCTION

Reinforcement learning(RL) involves training an agent to learn a policy that accomplishes a certain task in an environment. The objective of reinforcement learning is to maximize the expected future reward Sutton & Barto (1998) from a guiding signal. Mnih et al. (2015) showed that neural networks can be used to approximate state-action value functions used by an agent to perform discrete control based on a guiding reward. This was demonstrated in Atari games where the score was used as the reward signal. Similarly, continuous control of robotics arm was achieved by Lillicrap et al. (2016) minimizing the distance between end-effector and target location. Following these, other methods such as Schulman et al. (2017; 2015) were proposed to improve the sample efficiency of model-free algorithms with theoretical guarantees of policy improvement in each step. These algorithms assume that a guiding reward signal is available for the agent to learn optimal behavior for a certain task. However, in most cases of natural learning, such guiding signal is not present and learning is performed by imitating an expert behavior.

Imitation learning involves copying the behavior of an expert agent to accomplish the desired task. In the conventional imitation learning setting, a set of expert trajectories providing states and optimal actions $\tau = \{s_0, a_0, s_1, a_1, ..., s_n, a_n\}$ performed by an expert agent $\pi_E$ are available but the reward (or cost function), $r_E(s, a)$ used to achieve the expert behavior is not available. The goal is to learn a new policy $\pi$, which imitates the expert behavior by maximizing the likelihood of given demonstration trajectories.

A straightforward way for imitation learning is to direct learn the optimal action to perform given the current state as proposed by Pomerleau (1991); Duan et al. (2017). The policy $\pi$ can learn to imitate the expert behavior by maximizing likelihood of the condition distribution of action given states $p(a|s)$. This can be achieved by simply training a parameterized function (neural networks for instance) with state and action pairs from the expert trajectories. Since this involves end-to-end supervised learning, training is much more sample-efficient compared to reinforcement learning and overcomes inherent problems in model-free methods such as credit assignment(Sutton (1984)). However, since behavior cloning learns optimal action from a single state value only, it is unaware

of the future state distribution the current action will produce. Thus, errors are compounded in the future states leading to undesired agent behavior as shown by Ross et al. (2011); Ross & Bagnell (2010). Therefore, numerous training samples are required for behavior cloning to reduce errors in action prediction required for satisfactory imitation learning.

The second approach to imitation learning involves setting up exploration in a Markov Decision Process(MDP) setting. The goal then is to recover a reward signal that best explains the expert trajectories. Ng & Russell (2000) first introduced Inverse Reinforcement Learning(IRL), where the goal is to find a reward signal $\hat{r}$ from the trajectories such that the expert is uniquely optimal. After computing this estimated reward signal, usually, a model-free reinforcement learning performed to obtain the desired policy imitating the expert behavior by maximizing the expected discounted reward $\mathbb{E}_\pi(\sum_t \gamma^t \hat{r}(s_t, a_t))$. While this alleviates the problem of compounding errors as in behavior cloning, Ziebart et al. (2008) showed that estimating a unique reward function from state and action trajectories is an ill-posed problem.

Following the success of Generative Adversarial Networks(GANs) (Goodfellow et al. (2014)) in various fields of machine learning, adversarial learning has also been shown incorporated in the imitation learning framework. The recent work on Generative Adversarial Imitation Leaning or GAIL by Ho & Ermon (2016) showed that model-free reinforcement learning using the discriminator as a cost function can learn to imitate the expert agent with much less number of demonstrated trajectories compared to behavior cloning. Following the success of GAIL, there have extensions by Baram et al. (2017) to model-based generative imitation learning using a differentiable dynamics model of the environment. Robust imitation policy strategies using a combination of variational autoencoders (Kingma & Ba (2015); Rezende et al. (2014)) and GAIL has also been proposed by Wang et al. (2017).

The previous works assume that the expert trajectories consist of both action and state values from the optimal agent. However, optimal actions are usually not available in real-world imitation learning. For example, we often learn tasks like skipping, jump rope, gymnastics, etc. just by watching other expert humans perform the task. In this case, the optimal expert trajectories only consist of visual input, in other words, the consecutive states of the expert human with no action information. We learn to jump rope by trying to reproduce actions that result in state trajectories similar to the state trajectories observed from the expert. This requires exploring the environment in a structured fashion to learn the dynamics of the rope (for jump rope) which then enables executing optimal actions to imitate the expert behavior. The recent work of Liu et al. (2017) presents learning from observations only with focus to transferring skills learned from source domain to an unseen target domain, using rewards obtained by feature tracking for model-free reinforcement learning.

Inspired by the above method of learning in humans, we present a principled way of learning to imitate an expert from state information only, with no action information available. We first learn a distribution of the next state from the current state trajectory, used to estimate a heuristic reward signal enabling model-free exploration. The state, action and next states information from model-free exploration is used to learn a dynamics model of the environment. For the case of learning in humans, this is similar to performing actions for replicating the witnessed expert state trajectories, which in turn gives information about the dynamics of the environment. Once this forward model is learned, we try to find the action that maximizes the likelihood of next state. Since the forward model gives a function approximation for the environment dynamics, we can back propagate errors through it to perform model-based policy update by end to end supervised learning. We demonstrate that our proposed network can reach, with fewer iterations, the level close to an expert agent behavior (which is a pre-trained actor network or manually provided by humans), and compare it with reinforcement learning using a hand-crafted reward or a heuristics reward that is based on prediction error of next state learned from the optimal state trajectories of the expert.

## 1.1 NOTATIONS

We summarize the notations used in the paper in this section. Consider a Markov Decision Process (MDP) denoted as $(S, A, P, r, \rho_0, \gamma)$, where $S$ is the finite set of states, $A$ is the set of possible actions, $P : S \times A \to S$ is the transition probability distribution and $r : S \times A \to \mathbb{R}$ be the reward signal from state and actions, $\rho_0 \to \mathcal{R}$ is the initial state distribution and $\gamma \in (0, 1)$ is the discount factor. Let $\pi : S \times A \to (0, 1)$ be the policy that gives the conditional distribution of

actions given current state, $p(\boldsymbol{a}|\boldsymbol{s})$ and $R(\pi) = \mathbb{E}_\pi[\sum_t \gamma^t r_t(\boldsymbol{s}_t, \boldsymbol{a}_t)]$ is the discounted reward associated with the policy. We consider expert trajectories consisting of only optimal state distribution without any action information, $\tau_E = \{\boldsymbol{s}_0, \boldsymbol{s}_1, ..., \boldsymbol{s}_n\}$. The trajectories sampled from model-free exploration is denoted as, $\tau_{RL} = \{\boldsymbol{s}_0, \boldsymbol{a}_0, \boldsymbol{s}_1, \boldsymbol{a}_1, ..., \boldsymbol{s}_n, \boldsymbol{a}_n\}$. We use the terms dynamic model, state-transition probability and forward model interchangeably in the paper. $f_E(\boldsymbol{s}_t, \boldsymbol{a}_t)$ denotes the non-differentiable forward model of the environment and $f(\boldsymbol{s}_t, \boldsymbol{a}_t)$ denotes the differentiable version which is learned during the proposed training procedure. $\pi_{mf}$ denotes the model-free policy and $\pi_{mb}$ denotes the model-based policy network.

## 2 PROPOSED METHOD

While most previous works use trajectories containing both state and action information to infer a policy that imitates the expert behavior, our problem statement is to imitate an expert from optimal state trajectories only. This setting is common in many natural scenarios where only state information is available. For example, humans learning to swim by seeing videos or observing other experts swimmers, have access to the sensory stream of information containing states only. The optimal actions are obtained by trying to replicate the state trajectories in the real environment via exploration. In this work, we learn a time series predictor of next state given the current state. Subsequently, we estimate a heuristic reward signal at each step, based on the difference of predicted next state by the time series model and the actual next state taken by the policy network to learn a model-free policy with exploration. However, such heuristic methods suffer from the disadvantages of slow model-free training to reach satisfactory policy and provide no guarantees on convergence. Therefore we resort to model-based policy learning that learns a differentiable model of the environment dynamics used to learn policy by directly supervised learning. The proposed algorithm alternates between dynamic model parameters and policy parameters update using gradient propagation through the differentiable dynamics model. We formalize this setup in the following sections.

### 2.1 MODEL-FREE POLICY LEARNING VIA REWARD ESTIMATION

Following the work in Inverse Reinforcement Learning(IRL), it is possible to frame the imitation learning problem in the MDP framework. The imitation learning problem then reduces to finding a uniquely optimal reward signal that maximizes the likelihood of the observed demonstration trajectories. However, finding a uniquely optimal reward function, even with optimal trajectories containing both state and action information, is an ill-posed problem. The ill-posed nature of optimal reward estimation is further accentuated in the absence of action information in expert trajectories. As such, many solutions for parameterized families of reward estimation models maximizing the likelihood of only expert state trajectories might be sub-optimal to learn the desired task.

For model-free policy learning, we use a heuristic reward estimated by the error in next state prediction at each step. We make an assumption that estimating a locally optimal reward maximizing the likelihood of next step predicting, is intuitively globally optimal for learning the desired task. A straight-forward method to obtain such heuristic reward signal from the trajectories, $\tau_s$, is to learn a time series predictive model of the next state given the current state, $p(\boldsymbol{s}_{t+1}|\boldsymbol{s}_{1:t})$ from the expert state trajectories using time series modeling. In our case, we use an exponential of the difference between predicted states and actual next state associated with the action predicted by the policy network.

$$r_t(\boldsymbol{s}_t, \boldsymbol{a}_t) = \left. k \exp(-\frac{\|\hat{\boldsymbol{s}}_{t+1} - f_E(\boldsymbol{s}_t, \boldsymbol{a}_t)\|^2}{2\sigma^2}) \right|_{\hat{\boldsymbol{s}}_{t+1} \sim p(\boldsymbol{s}_{t+1}|\boldsymbol{s}_{1:t})} \tag{1}$$

where $k$ is a constant controlling gain of the reward signal and $\sigma$ controls the sensitivity of reward to divergence from the predicted state. This reward can be used for guiding any standard model-free reinforcement learning techniques to maximize the expected reward, $R(\pi_{mf}) = \mathbb{E}_{\pi_{mf}}[\sum_t \gamma^t r_t(\boldsymbol{s}_t, \boldsymbol{a}_t)]$. We assume on the intuition that locally optimal heuristic reward estimation will be sufficient to ensure global optimality for learning the desired task.

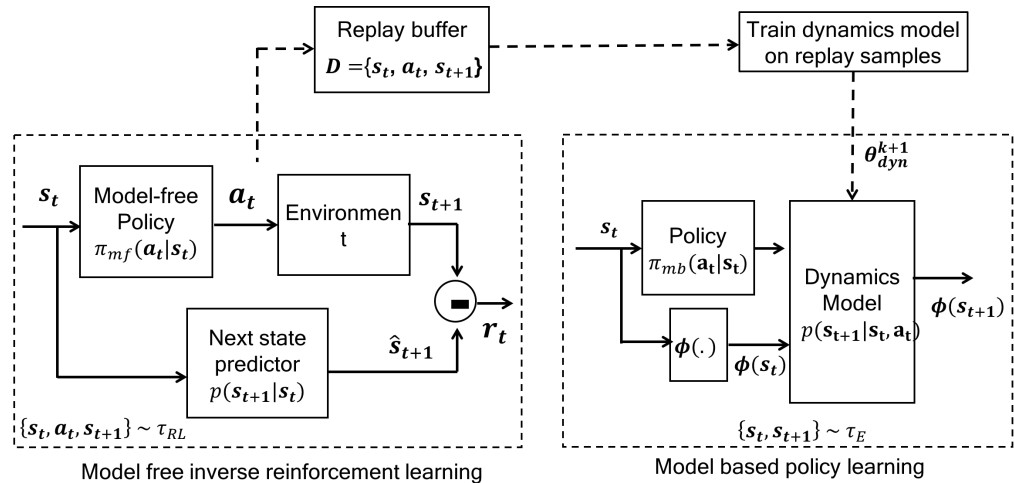

Figure 1: Showing the overall architecture of the proposed method. Firstly, model-free policy is updated using reward estimation from next state mismatch which storing the samples in replay buffer. These are used to update the differentiable dynamics model of the environment, which provides the error gradients for end-to-end model-based policy update from state trajectories

## 2.2 MODEL-BASED IMITATION LEARNING FROM STATE TRAJECTORIES

Consider there are $m$ sets of expert trajectory episode each consisting of $T$ states, given as $\tau_E = \{s_0, s_1, ..., s_n\}$, where $n = mT$. We assume that the trajectories in each episode are independent of each other. For the imitation learning problem, we wish to imitate an expert agent $\pi_E$ from state trajectories $\tau_E$. we formulate a maximum likelihood problem for state trajectories given a parameterized model of the agent policy, given as

$$\boldsymbol{\theta^*} = \arg\min_{\boldsymbol{\theta}} \Big[ -\sum_{j=1}^{n} \sum_{t=1}^{T} \log p(\boldsymbol{s_{t+1}} | \boldsymbol{s_{1:t}}; \boldsymbol{\theta}) \Big], \tag{2}$$

where $\boldsymbol{\theta}$ represents the parameter of the model. We assume that the random variables state($\boldsymbol{s_t}$), action($\boldsymbol{a_t}$) and next state $\boldsymbol{s_{t+1}}$ form a directed graphical model as shown in figure 2 (b). Following the natural dynamics of environments in reinforcement learning setting, we assume that control action for the agent $a_t$ is conditionally independent of other state and actions in the past given the current state. Distribution of the next state is conditionally independent of the other past states given current state and action following the MDP setting. In this framework, we can frame the model-based policy as an encoder network with action as the latent variable and the dynamics policy as the decoder network predicting the next state. The log-likelihood estimation loss, in this case, can be written as,

$$\mathcal{L}_{sas} = -\sum_{j=1}^{m} \sum_{t=1}^{T} \log \int_a p(\boldsymbol{s_{t+1}} | \boldsymbol{s_t}, \boldsymbol{a}; \boldsymbol{\theta_d}) p(\boldsymbol{a} | \boldsymbol{s_t}; \boldsymbol{\theta_e}) da, \tag{3}$$

where $\boldsymbol{\theta_e}, \boldsymbol{theta_d}$ are the encoder and decoder parameters respectively. Learning can be performed using $(\boldsymbol{s_t}, \boldsymbol{s_{t+1}})$ pairs from the expert trajectories by minimizing the above loss, sampling action values from the posteriori distribution $p(\boldsymbol{a_t} | \boldsymbol{s_t})$ using standard Markov Chain Monte Carlo (MCMC) methods or variational inference methods Kingma & Ba (2015); Rezende et al. (2014). However, the learned action from the encoder in this case will not mimic the actual control commands used to perform the desired task in the control environment.

We propose a constrained minimization cost function which enforces the decoder network to mimic the environment dynamics of the agent. The proposed cost function enforces that the decoders model minimizes the loss for dynamics model prediction of next state given the current state and action,

whereas the composition of encoder over the decoder minimizes prediction loss of the next state given the current state. This loss function is given as,

$$
\mathcal{L}_{proposed} = - \mathbb{E}_{(s_t, s_{t+1}) \sim \tau_E} \left( \log \int_a p(\boldsymbol{s_{t+1}} | \boldsymbol{s_t}, \boldsymbol{a}; \boldsymbol{\theta}_{dyn}) p(\boldsymbol{a} | \boldsymbol{s_t}; \boldsymbol{\theta}_{mb}) d\boldsymbol{a} \right)
$$
$$
- \mathbb{E}_{(s_t, a_t, s_{t+1}) \sim \tau_{RL}} \left( \log p(\boldsymbol{s_{t+1}} | \boldsymbol{s_t}, \boldsymbol{a_t}; \boldsymbol{\theta}_{dyn}) \right),
\tag{4}
$$

where we $\boldsymbol{\theta}_{dyn}$ are the parameters of dynamics model and $\boldsymbol{\theta}_{mb}$ are model-based policy network's parameters. Let us denote the first term of the loss term in equation 4 as model-based policy loss, $\mathcal{L}_{mb}$ and the second term is referred as dynamics model loss, $\mathcal{L}_{dyn}$. As shown in figure 1, we perform alternate minimizations on the proposed cost function and training on the encoder and decoder are performed on two separate datasets. Firstly, the dynamics model parameters are updated from the experience replay samples gathered during model-free exploration. Subsequently, the updated dynamics model is used as the decoder network in the above formulation with fixed weights while the model-based policy parameters are updated by the gradient, $\nabla_{\theta_{mb}} \mathcal{L}_{mb}$. This enforces the encoder network to act as a model-based policy that learns to predict the optimal action given the current state. During implementations, a deterministic encoder and decoder are used and a single action is sampled from the posterior distribution during training.

### 2.2.1 LEARNING DYNAMICS MODEL FROM MODEL-FREE SAMPLES

Since our expert trajectories only consist of sensory streams containing state information only, learning from the heuristic reward in a MDP setting can be slow and does not guarantee that the reward is optimal for the desired task. Thus we resort to dynamics model based learning which can propagate error gradients with respect to action information as discussed in the above section. Consider an analogy of a robot learning to navigate through a maze of expert state information only. It must first learn the state transition model $p(\boldsymbol{s_{t+1}} | \boldsymbol{s_t}, \boldsymbol{a_t})$ to navigate through the environment. Once dynamics model is learned, it can obtain the best action that takes the current state in optimal state trajectories to the next state. Solving for the desired action at each state ($p(a_t | s_t)$) is a maximum likelihood problem from the expert state trajectories which can be solved by end-to-end supervised learning.

Let us assume we have a parameterized model for agent dynamics, given as $\boldsymbol{s}_{t+1} = f(\boldsymbol{s_t}, \boldsymbol{a_t}; \boldsymbol{\theta}_{dyn})$, where $\boldsymbol{\theta}_{dyn}$ denotes parameters of the model. During model-free learning, we store the trajectories of $(\boldsymbol{s_t}, \boldsymbol{a_t}, \boldsymbol{s_{t+1}})$ that were encountered during exploration by the agent. Let us denotes the trajectories of these triplets as $\tau_{RL}$. For continuous state spaces, the gradient for dynamic model parameters are given as

$$
\nabla_{\theta_{dyn}} \mathcal{L}_{dyn} = \nabla_{\theta_{dyn}} \mathbb{E}_{(\boldsymbol{s_t}, \boldsymbol{a_t}, \boldsymbol{s_{t+1}}) \sim \tau_{RL}} \left[ \| f(\boldsymbol{s_t}, \boldsymbol{a_t}; \theta_{dyn}) - \boldsymbol{s_{t+1}} \|_2^2 \right]
\tag{5}
$$

which is gradient on the mean squared error loss between model predicted and true next state. For the discrete state space case, we can first maximize the probability of next state given the current state and action using a categorical loss. Any standard stochastic gradient descent algorithms can be used for the above optimizations. However, we use neural networks as function approximators which are shown to have approximation capacity for arbitrary non-linear function, although its non-convex nature of optimization does always guarantee a solution that is globally optimal. Recent techniques in stochastic gradient descent (Zeiler (2012); Kingma & Ba (2014) ) have alleviated this problem to a large extent.

If we assume the dynamics model is ideal in predicting the next state and there exists a unique action to reach the next state from the current state, then the proposed method is identical to behavior cloning, although true action information is not provided by the expert. Therefore, the performance of this method is upper bounded by the performance of behavior cloning model which learns from the true action information between states.

### 2.3 TRANSFORMING STATES BASED ON ACTION

In our formulation so far, the entire next state of the agent is predicted by the next state predictor. However, we found that predicting a part of the state which is dependent on action gives better

reward structure. This is in line with the work of Pathak et al. (2017), where the authors predict $\phi(s_t)$ as the latent representation of the neural network predicting the action from consecutive states, $a_t = g(\phi(s_t), \phi(s_{t+1}))$. This transformed state value, $\phi(s_t)$ is also used as the input and output for the dynamics model. This transformation is beneficial for two reasons: (i) It is difficult to learn the dynamics model for high dimensional state information and thus first projecting onto a low dimensional manifold to learn the dynamics model gives a more feasible learning framework. (ii) In case of transferring the learned dynamics model between different tasks that use the same environment, a common state input is required for the dynamics model, which can be achieved such transformation. In case of learning from videos, we use the agent position in the image frames as $\phi(s_t)$. For the case of linked arm reacher, we will use $phi(s_t)$ as the joint angles and joint velocities.

## 2.4 ALGORITHM

We now outline the algorithm based on the above discussed model-based policy learning framework.

---

**Algorithm 1** Model based imitation learning from state trajectories

---

1: **Input :** Given expert trajectories $\tau_E$, initial parameters $\boldsymbol{\theta}_{mb}^0$, $\boldsymbol{\theta}_{mf}^0$ and $\boldsymbol{\theta}_{dyn}^0$ and the state transformation $\phi(.)$
2: Learn next state predictor model
$$\boldsymbol{\theta_p} \leftarrow \arg\min_{\boldsymbol{\theta}_p} -\mathbb{E}_{\{s_{1:t+1}\}\sim\tau_E} \log p(\phi(\boldsymbol{s}_{t+1})|\boldsymbol{s}_{1:t}; \boldsymbol{\theta}_p)$$
3: **for** $k = 1, 2, 3, ...$ **do**
4:      Sample trajectory from model-free policy $\tau_k \in \pi_{mf}^k$ and add to the replay buffer $\tau_{RL}$
5:      Updated the dynamics model parameter using trajectories from the replay buffer
$$\boldsymbol{\theta_{dyn}^{k+1}} \leftarrow \arg\min_{\boldsymbol{\theta}_{dyn}^k} -\mathbb{E}_{(s_t,a_t,s_{t+1})\sim\tau_{RL}} \left( \log p(\phi(\boldsymbol{s_{t+1}})|\phi(\boldsymbol{s}_t), \boldsymbol{a}_t; \boldsymbol{\theta}_{dyn}^k) \right)$$
6:      Update model based parameter from expert trajectories with fixed dynamics model.
$$\boldsymbol{\theta}_{mb}^{k+1} \leftarrow \arg\min_{\boldsymbol{\theta}_{mb}^k} -\mathbb{E}_{(s_t,s_{t+1})\sim\tau_E} \left( \log \sum_a p(\phi(\boldsymbol{s_{t+1}})|\phi(\boldsymbol{s}_t), \boldsymbol{a}; \boldsymbol{\theta}_{dyn}^{k+1})p(\boldsymbol{a}|\boldsymbol{s}_t; \boldsymbol{\theta}_{mb}^k) \right)$$
7: **end for**

---

The above algorithm shows an iterative process where in each iteration we first train a model-free algorithm using the heuristic reward function. This step is necessary because we collect a certain amount of system dynamics data, $(s_t, a_t, s_{t+1})$, while training the model-free policy. Then, we train a system dynamics model using the above collected data. The action policy is then trained using the system dynamics model in the model-based part, which constitutes one cycle of the training. Subsequently, we repeat this cycle again starting from the model-free part. With each iteration, we collect additional system dynamics data, which results in a more precise dynamics model, leading to accurate action policy parameter gradients for updating the model-based policy.

The frequency of switching between model-free replay buffer collection and model-based update can be varied depending on the complexity of dynamics model. For state predictions from previous states, we used Long Short-Term Memory Network, proposed by Hochreiter & Schmidhuber (1997). For the policy and dynamics model, we use neural networks as function approximators. In this work, we assume that the state transformation $\phi(.)$ is manually specified in each experiments, although it is possible to learn such representation by learning a common transformation between states that predicts the action, $a_t = g(\phi(s_t), \phi(s_{t+1}))$.

## 3 EXPERIMENTAL RESULTS

We perform experimental evaluations on three kinds of environment, (i) Robotics arm reacher in 2d, (ii) Simple 2d obstacle avoidance based reacher, (iii) Learning to play simple games from raw video demonstrations. In each experiment, we show specific strengths of the proposed algorithm as follows.

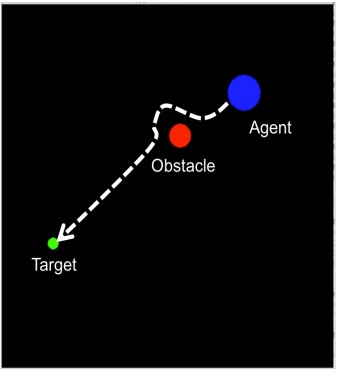 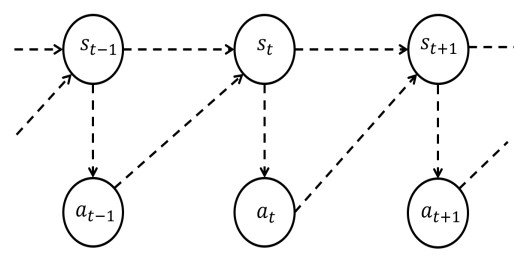

(a) Simple toy environment for obstacle avoidance.

(b) Directed graphical model showing the relation between next state, current state and action.

Figure 2: (a) Showing the toy obstacle avoidance environment used in our experiments, (b) The directed graphical model, which shows that next is dependent on current state and action, while current action is just dependent of the current state

### 3.1 ROBOTICS ARM REACHER IN 2D

We use roboschool reacher(OpenAI (2017); Brockman et al. (2016) environment to simulate two-link robotic arm that can move in two-dimensional space. The desired task is to learn reaching a given target from random starting configurations. The arm is controlled using angular torque values of both joints. We use state values consisting of angular position, angular velocity, the end effector location of the robotic link and the position of the target. The robotic arm angles and angular velocities were used as $\phi(\boldsymbol{s}_t)$, which is the portion of state dependent on action. In this experiment, we assume that the true reward function is known and in addition, we have some state trajectories from the expert policy. The goal is to leverage these existing state trajectories to learn a better policy in less number of steps. Reward signal, consisting of distance potential, electricity cost and the penalty for stuck joint, which is the default reward specified for the environment, was used. Specifically, we show 500 trajectories of optimal states each with 100 steps to learn model-based policy using the proposed method. We used neural networks with hidden layers containing $(128, 64)$ neurons for both model-based and model-free policies. Training was performed over 2000 episodes using Deep Deterministic Policy Gradients(DDPG) proposed by Lillicrap et al. (2016).

Figure 3(a) shows the comparison of proposed method against the DDPG algorithm. Our method learns the dynamics model from the model-free exploration, which quickly learns the simple environment dynamics, in this case, thereby learning an optimal policy imitating the state trajectories much faster than model-free training, which is shown in results. However, we found this is due to a large number of state trajectories that are shown to the proposed method. Since our performance is upper bounded by behavior cloning results, we share the same drawbacks of compounding errors and data-hungry policy learning.

### 3.2 2D OBSTACLE AVOIDER

In this experiment, we demonstrate that proposed algorithm can be used for direct end-to-end supervised imitation learning on novel tasks without resorting to model-free reinforcement learning. We refer to this setup as one-shot imitation learning, which we demonstrate on a simple toy environment. The environment is shown in figure 2(a). The environment consists of an agent which can freely move in 2D space. The agent is controlled by continuous position control where the action is the change in $(x, y)$ agent position at each time step. The goal is to reach the target location while avoiding the obstacle. Initially, we train our algorithm on an environment to avoid a single obstacle while reaching the target. We use state information as the absolute position of the agent, target and obstacle, the agent velocity and relative location of obstacle and target with respect to the agent. We use $\phi(\boldsymbol{s}_t)$ as the agent 2d position in the environment.

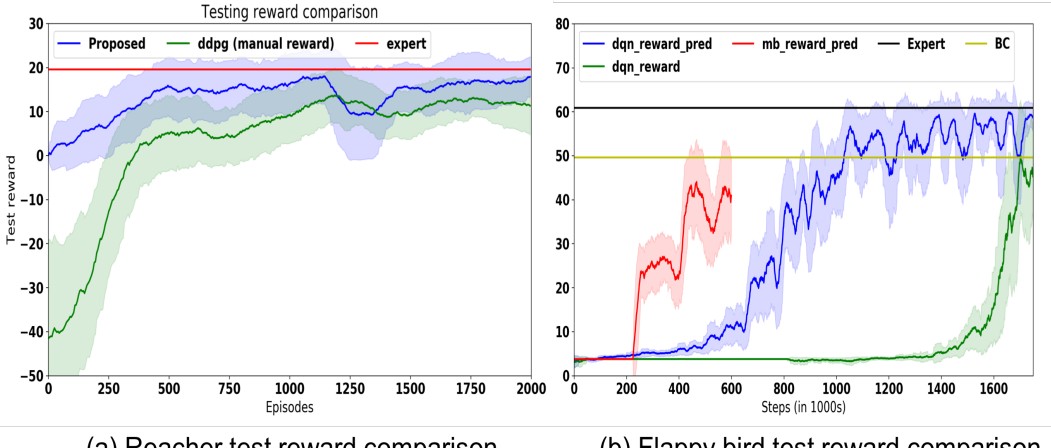

(a) Reacher test reward comparison      (b) Flappy bird test reward comparison

Figure 3: (a) Comparison of proposed method with model-free reinforcement learning methods, behavior cloning and proposed model-based method for Flappy birds. The model-based method is shown to surpass the model-free method in test reward performance with much less iterations, although being upper bounded by behavior cloning results (b) Comparison of the proposed method with continuous model-free methods on roboschool reacher showing better convergence performance with less iterations

For expert demonstration, we implemented a manually engineered policy that always avoids the obstacle. We used 1000 number of demonstrations containing only state trajectories to reach a target while avoiding a single obstacle. Out of 1000 demonstrations, 800 are used for training and 200 for validation. We first learn the time series prediction of the next state, used to compute the heuristic reward based using prediction error as discussed in section 2.1. For model-based policy, we use a MLP with $(64, 64)$ hidden units. We use the same policy network for both model-based and model-free policy. The dynamics model is also modeled as a neural network with a single hidden layer of 8 units for both state and action input. We used a switching frequency of 5 between the model-free and model-based updates. Using these setting for the proposed algorithm, we get a model-based policy and the dynamics model as output.

Using the dynamics model obtained from training with demonstrations of single obstacle avoidance, we perform one-shot imitation learning to learn avoidance of two obstacles. The algorithm is presented 500 samples of expert state trajectories for avoiding two obstacles. The model-based policy in the new setting is learned by step 6 of the proposed algorithm 1 using the previously learned dynamics model. Although the state information for the policy networks might change due to an additional obstacle, since $\phi(s_t)$, which is the agent 2d location, remains same in both cases, we can perform one-shot imitation learning in this case. We compare the results with respect to the expert policy and behavior cloning and report the average of test reward on 50 episodes. While the expert policy achieves average test reward of **3036**, and behavior cloning achieves **3939** and our imitation learning method gave a reward of **3805**. This demonstrates that our proposed algorithm can be used for one-shot imitation in environments with same dynamics and can produce comparable results to behavior cloning which was trained from true actions.

### 3.3 LEARNING POLICIES FROM RAW VIDEOS

In this experiment, we learn model-based control policies from raw pixels. We use the python reimplementation of the game Flappy bird (Lau (2017)). In this environment, the agent has to hover through the pipes without collision by flapping its wings. The environment has gravity and with each flap, the agent receives an upward acceleration which sends it to an upward parabolic trajectory. We choose this environment due to its complicated dynamics and show that our proposed method can learn to model this dynamics. We learn action policies from raw videos of just 10 episodes each with 1000 steps. The reward is assumed to be unknown in this case and we estimate the reward by the error in prediction of the next state as mentioned in section 2.1. The control action is a single discrete

command of whether to flap the bird's wings or not. We denotes this action space as $\{+1, -1\}$. For state information at each step we use 4 consecutive frames resized to $(80 \times 80 \times 4)$. We also assume that the absolute position of the bird's location is available to us, which we use as $\phi(s_t)$. This can be also computed by learning a simple object detector from image frames.

For the next state predictor, we use an LSTM predictor that outputs the next position of the agent location given the sequence of states witnessed so far. The model-free reward prediction step receives a reward signal based on the difference of the actual next state taken the policy, from the predicted next state by LSTM. This reward is used to train DQN(Mnih et al. (2015)) for model-free policy update which collects data to train the dynamics model, which in turn trains the model-based policy network. We also train vanilla DQN using a standard reward, which in this case is 0.1 for each time step and +1 reward if the agent can successfully pass through a pipe. To compare the various methods, we used the reward used by vanilla DQN as a baseline for comparison.

For the model-based policy we used a convolutional neural network (CNN) with soft-max output, which is essentially a binary classification task. For the DQN model-free policy, the last layer predicts the q-values and therefore has linearly activation. In this case, we first learn dynamics model, which is approximated as a multi layered perceptron (MLP) with single hidden layer of 16 units for both state and action inputs. It learns to regress the next state from the current state and actions minimizing mean squared error loss. After this, from the expert demonstration of state trajectory($\tau_E$), we find the current optimal action that minimizes the next state $a_k^* = \arg\min_a \|p(s_{t+1} - f(s_t, a_t)\|\|_{a_t \in \{0,1\}, \{s_{t+1}, s_t\} \sim \tau_E}$. The next step is to find the model-based policy by behavior cloning on the states ($s_t \sim \tau_E$) and current optimal actions, $a_k^*$. We found that, the number of +1(flap) actions by the agent are far less frequent compared to the number of -1(no flap) actions which cause an unbalance in distribution. Thus, we used class balancing techniques for learning both model based policy update and direct behavior cloning from true actions (baseline) results.

Figure 3(b) shows the comparison of proposed model-based method with estimated reward (*mb_reward_pred*), behavior cloning (*bc*), model-free RL using DQN with standard known reward (*dqn_reward*) and with estimated reward (*dqn_reward_pred*). It is to be noted that although original reward just provides a constant reward (of 0.1 and +1 bonus) without constant guidance at each step value, estimated reward provides a dense guidance signal which leads to faster convergence as shown in the comparison results. We take desired samples from the model-free training, based on prioritized sampling technique with regards to the estimated reward signal, to perform model-based policy update. We found that prioritized sampling was essential to learn a good dynamics model of the environment. The results show that the model-based policy learns behavior close to optimal in much fewer steps compared to the model-free counterparts. However, its performance in upper bounded by the behavior cloning method which achieves an average test reward (over 20 iterations) of **49.62**. The expert score is **60.9**. Since DQN with estimated reward signal learns via exploration in a MDP setting, it can surpass the performance of behavior cloning since the number of expert demonstrations is limited in this case.

## 4 CONCLUSION

We presented a model-based imitation learning method that can learn to act from expert state trajectories in the absence of action information. Our method uses trajectories sampled from the model-free policy exploration to train a dynamics model of the environment. As model-free policy is enriched through time, the forward model can better approximate the actual environment dynamics, which leads to improved gradient flow, leading to better model-based policy update which is trained in a supervised fashion from expert state trajectories. In the ideal case, when dynamics model perfectly approximates the environment, our proposed method is equivalent to behavior cloning, even in the absence of action information. We demonstrate that the proposed method learns the desired policy in less number of iterations compared conventional model-free methods. We also show that once the dynamics model is trained it can be used to transfer learning for other tasks in a similar environment in an end-to-end supervised manner. Future work includes tighter integration of the model-based learning and the model-free learning for higher data efficiency by sharing information (1) between the model-free policy $\pi_{mf}$ and the model-based policy $\pi_{mb}$ and (2) between the next state predictor $p(s_{t+1}|s_t)$ and the dynamics model $p(s_{t+1}|s_t, a_t)$ and (3) improving the limitations

of compounding errors and requirement of large number of demonstration, by adversarial training which can maximize likelihood of future state distributions as well.

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
