# OpenReview forum: "Model-based imitation learning from state trajectories"
_ICLR.cc/2018/Conference — Reject_

### Official Review · AnonReviewer2 · 2017-11-26
**Addresses an important problem, but misses existing research and results are unconvincing**

**Rating:** 4
**Confidence:** 4

**Review:**

The problem addressed here is imitation learning when no action information is available, which is an important problem in robotics for instance. The main idea of the proposed method is to produce a policy that matches the states observed in the expert trajectories, and this is achieved via a somewhat complex mix of model-free and model-based learning.

My main issues with the paper are:
- It does not cite or discuss a very important piece of related work: "Imitation from Observation: Learning to Imitate Behaviors from Raw Video via Context Translation" (Liu et al., 2017)
- The empirical results are unconvincing - it seems like in all problems they use there is a straightforward mapping from state feature differences to actions, as pointed out in an anonymous comment.

Additionally, it would have been nice to show empirically how helpful the model-based component of their approach is.

---

> ### Author Response · Authors · 2017-12-22
> **Rebuttal**
>
> We thank the reviewer for the overall constructive comments.
>
> Q : It does not cite or discuss a very important piece of related work: "Imitation from Observation: Learning to Imitate Behaviors from Raw Video via Context Translation" (Liu et al., 2017)
> A: Thank you for pointing out the relevant prior work using observations only. We added citation to this work in the introduction section of the paper. Our main contribution in this work is to show that the proposed method uses a combination of model-based and model-free methods for acceleration in imitation learning from observations alone. Although the mentioned prior work is similar to the proposed method, it's main focus is on transferring learned tasks on expert observations in a source domain to a novel target domain.
>
> Q: The empirical results are unconvincing - it seems like in all problems they use there is a straightforward mapping from state feature differences to actions, as pointed out in an anonymous comment.
> A : We agree that our experiments are simple in nature, with easy to learn dynamics model, which is a drawback of the current evaluation scheme. However, the main contribution of this paper is to present the novel idea that combination of proposed model-based and model-free training has the advantage of accelerated training for imitation learning from observation alone, which can be illustrated by these simple setups. In the future, we plan to build upon the current idea on complex dynamics model setup as well for future work.

---

### Official Review · AnonReviewer1 · 2017-11-26
**Interesting argument for model-based imitation learning**

**Rating:** 7
**Confidence:** 3

**Review:**

Model-Based Imitation Learning from State Trajectories

SIGNIFICANCE AND ORIGINALITY:

The authors propose a model-based method for accelerating the learning of a policy
by observing only the state transitions of an expert trace.
This is an important problem in many fields such as robotics where
finding a feasible policy is hard using pure RL methods.

The authors propose a unique two step method to find a high-quality model-based policy.

First: To create the environment model for the model-based learner,
 they need a source of state transitions with actions ( St, At,xa St+1 ).
To generate these samples, they first employ a model-free algorithm.
The model-free algorithm is trained to try to duplicate the expert state at each trajectory.
In continuous domains, the state is not unique … so they build a soft next state predictor
that gives a probability over next states favoring those demonstrated by the expert.
Since the transitions were generated by the agent acting in the environment,
these transitions have both states and actions ( St, At, St+1 ).
These are added to a pool.

The authors argue that the policy found by this model-free learner is
not highly accurate or guaranteed to converge, but presumably is good at
generating transitions relevant to the expert’s policy.
(Perhaps slowly reducing the \sigma in the reward would improve accuracy?)
I guess if expert trace data is sparse, the model-free learner can generate a lot
of transitions which enable it to create accurate dynamics models which in turn
allow it to extract more information out of sparse expert traces?

Second: They then train a model based agent using the collected transitions ( St, At, St+1 ).
They formulate the problem as a maximum likelihood problem with two terms:
an action dynamics model which is learned from local exploration using the learner’s own actions and outcomes
and expert policy model in terms of the actions learned above
that maximizes the probability of the observed expert’s trajectory.
This is a nice clean formulation that integrates the two processes.
I thought the comparison to an encoder - decoder network was interesting.

The authors do a good job of positioning the work in the context of recent work in IML.

It looks like the authors extract position information from flappy bird frames,
so the algorithm is only using images for obstacle reasoning?


QUALITY

The propose model is described fairly completely and evaluated on
a “reaching" problem and the "flappy bird” game domain.
The evaluation framework is described in enough detail to replicate the results.

Interestingly, the assisted method starts off much higher in the “reacher” task.
Presumably this task is easy to observe the correct actions.

The flappy bird test shows off the difference between unassisted learning (DQN),
model free learning with the heuristic reward (DQN+reward prediction)
and model based learning.

Interestingly, DQN + heuristic reward approaches expert performance
while behavioral cloning never achieves expert performance level even though it has actions.

Why does the model-based method only run to 600 steps and stopped before convergence??
Does it not converge to expert level?? If so, this would be useful to know.

There are minor grammatical mistakes that can be corrected.

After equation 5, the authors suggest categorical loss for discrete problems,
but cross-entropy loss might work better. Maybe this is what they meant.


CLARITY

The overall approach and algorithms are described fairly clearly. Some minor typos here and there.

Algorithm 1 does not make clear the relationship between the model learned in step 2 and the algorithms in steps 4 to 6.

I would reverse the order of a few things to align with a right to left ordering principle.
In Figure 1, put the model free transition generator on the left and the model-based sample consumer on the right.
In Figure 3, put the “reacher” test on the left and the “flappy bird” on the right.


PROS AND CONS

Interesting idea for learning quickly from small numbers of samples of expert state trajectories.

Not clear that method converges on all problems.

Not clear that the method is able to extract the state from video — authors had to extract position manually
(this point is more about their deep architecture than the imitation framework they describe -
though perhaps a key argument for the authors is the ability to work with small numbers of
expert samples and still be able to train deep methods ) ??


POST REVIEW SUBMISSION:

The authors make a number of clarifying comments to improve the text and add the reference suggested by another reviewer.

---

> ### Author Response · Authors · 2017-12-22
> **Rebuttal**
>
> Thank you for the overall encouraging review. We address some of the concerns in the following,
>
> Q : Not clear that method converges on all problems.
> A: Yes it does not converge on all dynamics models. Currently, the main drawback of the method is that it cannot model complex dynamics models like raw video transitions as mentioned in the anonymous comment also.
>
> Q : Not clear that the method is able to extract the state from video — authors had to extract position manually
> A: Learning the useful state representations from raw video is a challenging problem. In literature, Pathak. et al. ICML 2017 proposes to use a feature extractor \phi, which learns to predict the action given the current and next state. However, in our case, we simplify the assumption by manually specifying parts of the state that depends on the actions. This is a limitation of the proposed method but we hope to address this issue in the future versions using methods in literature, such as Pathak. et al. ICML 2017.
>
> The overall approach and algorithms are described fairly clearly. Some minor typos here and there.
> A : We changed the typos and reordered the figures.

---

### Official Review · AnonReviewer3 · 2017-11-30
**In summary, this is a poorly written paper that seems to rely on a lot of heuristics that are not well motivated. Also the results are not convincing. Clear reject.**

**Rating:** 3
**Confidence:** 5

**Review:**

The paper presents a model-based imitation learning framework which learns the state transition distribution of the expert. A model-based policy is learned that should matches the expert transition dynamics. The approach can be used for imitation learning when the actions of the expert are not observed, but only the state transitions (which is an important special case).

Pros:
- The paper concentrates on an interesting special case of imitation learning

Cons:
- The paper is written very confusingly and hard to understand. The algorithm needs to be better motivated and explained and the paper needs proof reading.
- The algorithm is based on many heuristics that are not well motivated.
- The algorithm is only optimizing the one step error function for imitation learning but not the long term behavior. It heavily relies on the learned transition dynamics of the expert p(s_t+1|s_t). This transition model will be wrong if we go away from the expert's trajectories. Hence, I do not see why we should use p(s_t+1|s_t) to define the reward function. It does not prevent the single step
errors of the policy to accumulate (which is the main goal of inverse reinforcement learning)
- The results are not convincing
- Other algorithms (such as GAIL) could be used in the same setup (no action observations). Comparisons to other imitation learning approaches are needed.

In summary, this is a poorly written paper that seems to rely on a lot of heuristics that are not well motivated. Also the results are not convincing. Clear reject.


More detailed comments
- It is unclear why a model-based and model-free policy need to be used. Is the model-based policy used at any time in the algorithm? If it is just used as final result, why train it iteratively? Why can we not just also use the model-based policy for data collection?
- It is unclear why the heuristic reward function makes sense. First of all, the defined reward is stochastic as \hat{s}_t+1 is a sample from the next state from the expert's transition model. Why do not we use the mean of the transition model here, then it would not be stochastic any more. Second, a much simpler reward could be used that essentially does the same thing. Instead of requiring a learned dynamics model f_E for predicting the next state, we can just use the experienced next state s_t+1. Note that the reward function for time step t can depend on s_t+1 in an MDP.
- The objective that is optimized (Eq. 4) is not well defined. A function is not an objective function if we can only optimize part of it for theta while keeping theta fixed for the other part. It is unclear which objective the real algorithm optimizes
- There are quite a few confusions in terms of notation. Sometimes, a stochastic transition model p(s_t+1|s_t, a_t) is used and sometimes a deterministic model f_E(s,a). It is unclear how they relate.
- Many other imitation learning techniques could be used in this setup including max-entropy inverse RL [1], IRL by distribution matching [2] and the approach given in [3] and GAIL. A comparison to at least a subset of these methods is needed

[1] B. Ziebart et al, Maximum Entropy Inverse Reinforcement Learning, AAAI 2008
[2] Arenz, O.; Abdulsamad, H.; Neumann, G. (2016). Optimal Control and Inverse Optimal Control by Distribution Matching, Proceedings of the International Conference on Intelligent Robots and Systems (IROS)
[3] P Englert, A Paraschos, J Peters, MP Deisenroth, Model-based Imitation Learning by Probabilistic Trajectory Matching, IEEE International Conference on Robotics and Automation

---

### Public Comment · (anonymous) · 2017-11-09
**Simple baseline**

I think there is an important baseline missing, namely:
- Train a model to predict the action a_t using (phi(s_t), phi(s_{t+1})) as input, using samples obtained from the model-free policy.
- Use this model to predict actions performed in the expert trajectories.
- Train a standard imitation learner using the expert state trajectories together with the predicted actions as targets.

This should perform similarly to behavior cloning using true actions if the learned action predictor is accurate, which I think should be the case due to the way states are represented. For example, if I understand correctly in the 2D Obstacle Avoider task the action is simply a_t = phi(s_{t+1}) - phi(s_t). In Flappy Bird, the action is 1 if phi(s_{t+1})-phi(s_t) > 0 and -1 otherwise. It should be possible to learn both of these action predictors using very few samples.

This baseline would have more trouble if the inputs were videos and the hardcoded phi(s_t) was not provided, because the classifier would receive a high-dimensional input and would need more samples with known actions to fit its parameters. Did you try your method on Flappy Bird using only video, without providing the phi(s_t) as input?

---

> ### Author Response · Authors · 2017-12-22
> **Alternative simple baseline**
>
> Thank you for illustrating an alternative model-based method. We believe this is a useful baseline method that we should compare our method against. We agree that since our experiments are simple in nature, this proposed alternative method might perform equally well compared to the proposed method.
>
> However, the main difference between the suggested method and the proposed method, is that, the suggested method attempts to learn the inverse dynamics of the system (eg. given two locations (a,b) in space of the end-effector, find the torque value for moving a robotic arm from a to b) which might be difficult to learn in a general setting. Our proposed method learns the forward dynamics (eg. given current locations 'a' of the end-effector and torque values, find the next location), which might have a well-defined equation in mechanics in most general cases. However, we do agree that for the simple experimental evaluations that we have performed both, inverse and forward dynamics might be of equal difficulty.
>
> No, in our case we manually specify the location of the flappy-bird as \phi(s_t). It is challenging to directly learn dynamics model between raw video streams as has been already pointed out in the comment, and it is a limitation of the current proposed method. One method can be to automatically learn \phi(s_t) in case of high dimensional inputs, using action prediction from consecutive states, using ideas in prior methods, like Pathak. et al. ICML 2017.

---

### Decision · Program_Chairs · 2018-01-29
**ICLR 2018 Conference Acceptance Decision**

**Decision:**

Reject

**Comment:**

The paper is hard to follow at times. The heuristic reward has little justification -- not clear how
this would extend to other domains. Lack of empirical comparisons (see e.g. Hester et al., Deep Q-Learning from Demonstrations, 2017).